# Reg3α and Reg3β Expressions Followed by JAK2/STAT3 Activation Play a Pivotal Role in the Acceleration of Liver Hypertrophy in a Rat ALPPS Model

**DOI:** 10.3390/ijms21114077

**Published:** 2020-06-07

**Authors:** Naohiko Otsuka, Masato Yoshioka, Yuki Abe, Yasuhiko Nakagawa, Hiroshi Uchinami, Yuzo Yamamoto

**Affiliations:** 1Department of Gastroenterological Surgery, Akita University Graduate School of Medicine, Akita 0108543, Japan; n.otsuka0719@gmail.com (N.O.); yuki-a@gipc.akita-u.ac.jp (Y.A.); y-nakagawa@gipc.akita-u.ac.jp (Y.N.); huchi@gipc.akita-u.ac.jp (H.U.); 2Department of Nursing, Akita University Graduate School of Health Science, Akita 0108543, Japan; masato@gipc.akita-u.ac.jp

**Keywords:** ALPPS, Reg3α, Reg3β, JAK2/STAT3 pathway, liver hypertrophy

## Abstract

To explore the underlying mechanism of rapid liver hypertrophy by liver partition in associating liver partition and portal vein ligation for staged hepatectomy (ALPPS), liver partition at different sites was investigated. Increased inflammatory cytokines owing to the liver partition have been reportedly responsible. If this were true, rapid liver hypertrophy should be achieved regardless of where the liver was split. A male Sprague-Dawley rat model was created, in which a liver split was placed inside the portal vein ligated lobe (PiLL), in addition to the ALPPS and portal vein ligation (PVL) models. Liver regeneration rate, inflammatory cytokine levels, activation status of the Janus kinase 2/signal transducer and activator of transcription 3 (JAK2/STAT3) pathway and expressions of regenerating islet-derived (Reg)3α and Reg3β were investigated. The liver regeneration rate was significantly higher in the ALPPS group than in the PiLL group, whereas inflammatory cytokine levels were nearly equal. Additional volume increase in ALPPS group over PVL and PiLL groups was JAK2/STAT3-dependent. Reg3α and Reg3β expressions were observed only in the ALPPS group. An increase in inflammatory cytokines was not enough to describe the mechanism of rapid liver hypertrophy in ALPPS. Expressions of Reg3α and Reg3β could play an important role in conjunction with an activation of the JAK2/STAT3 pathway.

## 1. Introduction

Patients with large or multiple liver tumors require extended liver resection. However, postoperative liver failure may occur in these patients due to insufficiency of remnant liver volume [1,2]. To solve this problem, portal vein ligation (PVL) or embolization is performed preoperatively, and the future liver remnant (FLR) increases by 10% to 45% with a waiting period from two to eight weeks [3,4,5]. However, tumor progression during this period is an issue, and the rate of liver resection after portal vein occlusion remains at 70% [3,4].

As a new strategy, associating liver partition and portal vein ligation for staged hepatectomy (ALPPS) was introduced [5]. Although the ALPPS procedure consists of step 1 (PVL and split of the FLR) and step 2 (hepatectomy), a rapid liver regeneration during step 1 is characteristic in comparison to PVL alone. This method enables the FLR to increase by 70% to 80% within only 10 days [5,6]. Although a high mortality (90-day mortality of 9%) and morbidity (Grade ≥ IIIb of 40% in the Clavien-Dindo classification) becomes a problem [7], technical modifications such as partial or mini-ALPPS have been reported and contribute to fewer complications [8,9]. Although many studies on ALPPS have been reported in clinical settings, there are not many works on basic research, and therefore the mechanism of rapid liver hypertrophy remains unclear. There are only several animal experiments trying to explore the underlying mechanism by comparing the ALPPS model with the PVL model [10,11,12,13]. What their findings suggested in common was that the increase in inflammatory cytokines, owing to the liver partition, was responsible. However, if the increased inflammatory cytokines alone were the element that characterizes rapid liver hypertrophy in ALPPS, a liver partition placed at a different site from the original ALPPS should also reproduce similar rapid hypertrophy, because wherever the liver partition is placed, destruction of liver parenchyma itself would promote the release of inflammatory cytokines [10,11,12,13]. In this study, we focused on the dramatic liver regeneration during step 1 of the ALPPS procedure, and first investigated whether the liver partition at different sites had an influence on the subsequent speed of liver hypertrophy and how inflammatory cytokines were involved. Secondly, we explored a novel factor related to this mechanism.

## 2. Results

### 2.1. Effects of the Site of the Liver Partition on Liver Hypertrophy

Rats were allocated into three groups (Figure 1). As in previously published studies [10,11,12,13], we prepared two groups to investigate the role of liver parenchymal partition over PVL: (a) PVL group, all branches of the portal vein were ligated except the branch to the right median lobe (RML); and (b) ALPPS group, the portal vein branches were ligated as in the PVL group and the RML was split along the right side of the demarcation line that emerged after ligating the left branch of the portal vein [11]. To investigate whether the site of the liver partition placed in ALPPS has any significant role, we created an additional experimental group in which the liver partition was intentionally placed inside the lobe where the portal venous flow had been occluded: (c) Partition inside the Ligated Lobe (PiLL) group, 80% portal vein ligation and liver partition inside the left lateral lobe (LLL). In all three groups, arterial branches to all liver lobes were preserved so that the arterial blood perfusion of the portal vein ligated lobe could be maintained. Hematoxylin-eosin staining of the liver tissues from the LLL on day 1 in every group showed neither ischemic changes nor apoptosis, affirming that the blood perfusion was preserved in the portal vein ligated lobe as well.

The weight of the RML of the liver to body weight (BW) ratio increased over time in all groups. However, there was a significant difference in the trend between the groups on postoperative days 1, 3 and 7 (Figure 2a). As expected, the RML/BW ratio (%) in the ALPPS group was significantly higher than in the PVL group. In particular, the regeneration speed in the ALPPS group was significantly faster than the PVL group until day 1, and after that, the difference in the RML/BW ratio that occurred during this period was just maintained, suggesting that immediate response within one day after the step 1 intervention served as a key of rapidity of regeneration in ALPPS. Interestingly, the PiLL group displayed a significantly lower RML/BW ratio than the ALPPS group and not different from the PVL group. In agreement with this profile of the RML/BW ratio, the proliferating cell nuclear antigen (PCNA) labeling index on day 1 was significantly higher in the ALPPS group than in the other two groups, showing that more hepatocytes entered into the cell-division cycle (PCNA mainly detects the cells in late G1 and S phases) in the ALPPS group; there was no difference between the PVL and PiLL groups (Figure 2b). In addition to PCNA staining, we stained the liver tissue on days 1 and 3 for phosphorylated Histone H3 (pH3), a more specific biomarker that stains cells in late G2 and mitosis (Figure 2c). The ALPPS group displayed much more pH3-positive hepatocyte nuclei than the PVL and PiLL groups on day 1. On day 3, in the PVL and PiLL groups as well, the number of pH3-positive hepatocyte nuclei had shown a fair increase belatedly, catching up with that in the ALPPS group. Although a delayed increase of the PCNA labelling index and pH3 staining in the PVL and PiLL groups eliminated the dominance in the ALPPS group by day 3, considering the specific feature of the ALPPS strategy—early rapid volume increase, like a boost-start—these changes over time seemed natural. It is very meaningful that more hepatocytes enter the cell cycle significantly earlier after ALPPS than after PVL or PiLL.

Aspartate aminotransferase (AST), alanine aminotransferase (ALT) and lactate dehydrogenase (LDH) showed the highest values on day 1 in all groups, and gradually decreased thereafter (Figure 3). There were no significant differences among the three groups regarding serum levels of liver enzymes.

### 2.2. Effects of the Site of the Liver Partition on Cytokine Production

Since the RML/BW ratio of the PiLL group did not increase much, remaining rather the same as the PVL group despite splitting of the liver parenchyma, we were prompted to measure the inflammatory cytokines interleukin-6 (IL-6) and tumor necrosis factor-α (TNF-α).

Figure 4a illustrates the relative increase in messenger ribonucleic acid (mRNA) expressions of *IL-6* and *TNF-α* on day 1 over their constitutive expressions. Even in the PVL group, *IL-6* gene was strongly up-regulated by eight times (8.66 ± 10.6). However, the degree of up-regulation in the ALPPS and PiLL groups was tremendous and extended to about 30 times (34.0 ± 7.30 in ALPPS, 27.8 ± 10.3 in PiLL). This result, as predicted, confirms that expression of *IL-6* gene was vigorously enhanced also in the PiLL group, to the same extent as the ALPPS group. Although the scale was different, the same trend was reproduced in *TNF-α* gene. In line with this, serum concentrations of these cytokines showed similar profiles (Figure 4b). When comparing profiles of cytokine production with those of the RML/BW ratio, a simple deduction that the increase of IL-6 or TNF-α owing to the parenchymal partition is a pivotal element which characterizes the rapid volume increase in ALPPS seemed to be imprudent. Looking only at the differences between the ALPPS and PVL groups, the increase of inflammatory cytokines might appear to be a specific feature of the ALPPS group, as was the case in previous reports. Whereas, comparing the ALPPS group with the PiLL group, the difference in the RML/BW ratio could not be explained by cytokine concentrations alone.

### 2.3. Expression of Hepatocyte Growth Factor (HGF) mRNA in the Liver Tissue of FLR

Since it is well known that there is a close relationship between IL-6 and HGF expression during the period of step 1 of the ALPPS procedure [13], we compared expression levels of *HGF* mRNA among the three groups (Figure 5). As expected from the serum concentration of IL-6, *HGF* expression was more intense in the ALPPS and PiLL groups than in the PVL group, and there was no difference between the ALPPS group and the PiLL group. This result once again convinced us that the orthodox regenerative stimuli, like a cascade from IL-6 through HGF to cell division, alone were not sufficient to explain the boost-start mechanism of liver regeneration by day 1 after intervention.

### 2.4. Janus Kinase (JAK) 2/Signal Transducer and Activator of Transcription (STAT) 3 Pathway in the ALPPS Group

Activation of the JAK2/STAT3 pathway is reportedly the main avenue from IL-6 signaling to cell proliferation [14,15]. To gain insight into the regulatory commitment of the JAK2/STAT3 pathway in enhancing more liver regeneration in the ALPPS group than the PVL and PiLL groups, we blocked the JAK2/STAT3 pathway with JAK2 inhibitor AG490. Rats were sacrificed on day 1, and RML/BW ratio, mRNA expression of *PCNA* and PCNA labeling index were examined. Unexpectedly, in the PVL and PiLL groups, there was no influence of JAK2 inhibition on the RML/BW ratio. Whereas, in the ALPPS group, the RML/BW ratio was significantly reduced, suppressed to the same level as the PVL and PiLL groups (Figure 6a). In other words, inhibition of the JAK2/STAT3 pathway abolished only an additional increase of the liver volume in the ALPPS group over the PVL and PiLL groups. Parallelly, *PCNA* mRNA expression and PCNA labeling index in the ALPPS group were also suppressed to the level observed in the PVL and PiLL groups (Figure 6b,c). These results suggested that the additional volume increase in the ALPPS group was JAK2/STAT3 pathway-dependent and that the basal increase of the volume in the PVL and PiLL groups was not dependent on the JAK2/STAT3 pathway, or not fully at least.

To confirm the JAK2/STAT3 pathway dependency of the ALPPS specific volume increase, activation of STAT3 was investigated by immunohistochemistry using anti-phosphorylated STAT3 (pSTAT3) antibody. Of interest, pSTAT3 was not stained at all in the PVL and PiLL groups despite high IL-6 concentrations, confirming that the JAK2/STAT3 pathway is not activated in these groups (Figure 7a,c). On the other hand, the ALPPS group displayed a strong staining for pSTAT3 in the nuclei of hepatocytes (Figure 7b). These results were in good agreement with the study using the JAK2 inhibitor. Even more interestingly, staining intensity of pSTAT3 was stronger in the area closer to the liver partition than the area away from it (Figure 7d). Taken together, activation of the JAK2/STAT3 pathway plays a crucial role in an additional volume increase in the ALPPS group, and is independent of inflammatory cytokines.

### 2.5. Regenerating Islet-Derived (Reg)3α and Reg3β Are Related to Rapid Liver Hypertrophy in ALPPS

To explore a factor that activates the JAK2/STAT3 pathway in the ALPPS group, we performed comprehensive gene analysis by a complementary RNA (cRNA) microarray system. Of 20,000 genes analyzed, approximately 100 genes were up-regulated in the ALPPS group relative to the PVL group (Table 1). Among them, we focused on *Reg3α* and *Reg3β* genes for further investigation because they are reportedly strongly related to cell proliferation and differentiation through the JAK2/STAT3 pathway [16,17].

Figure 8a shows mRNA expressions of *Reg3α* and *Reg3β* in the liver tissue from the sham operation group. *Reg3α* mRNA was constitutively expressed, but *Reg3β* mRNA was negative or at least very faint. We evaluated *Reg3α* and *Reg3β* mRNAs as a relative expression to *glyceraldehyde 3-phosphate dehydrogenase* (*GAPDH*) on day 1. Actually, expression was more prominent in the ALPPS group than in the PVL group (Figure 8b,c). More importantly, the *Reg3β* gene was barely expressed in either the PVL or PiLL group. The ALPPS group alone displayed an up-regulation of *Reg3β* gene by more than two times (Figure 8c). This means that the expression profile of the *Reg3β* gene is parallel to that of the RML/BW ratio, the PCNA labeling index, as well as the activation of the JAK2/STAT3 pathway. It was also noteworthy that despite the increase of IL-6, little expression of the *Reg3β* gene could be observed in the PiLL group. As for *Reg3α*, although the difference did not reach statistical significance, it was less expressed in the PiLL group than in the ALPPS group, demonstrating a similar profile to *Reg3β* (Figure 8b). These features of *Reg3α* and *Reg3β* gene expression, along with activation of the JAK2/STAT3 pathway, were more convincing as an element characterizing ALPPS than an increase in inflammatory cytokines.

### 2.6. Location of Reg3α and Reg3β Protein Expression

Since mRNA expression is not always correlated with protein expression, we confirmed protein expression by immunohistochemistry on day-1 tissues. Consistent with mRNA expression, a constitutive weak staining of Reg3α protein was detected in the sham liver (Figure 9a), and the staining intensity in the PVL and PiLL groups was not different from the sham liver (Figure 9b,d). In the ALPPS group, the intensity was stronger than these groups (Figure 9c), and the expression was diffuse throughout the liver (Figure 9e). Regarding the Reg3β protein, the difference in expression among the groups was very distinctive. It was very strongly expressed in the ALPPS group (Figure 9h), whereas it was completely negative in the sham, PVL and PiLL groups (Figure 9f,g,i). Furthermore, the induction of Reg3β protein was more prominent in zone 3 of hepatic lobules closer to the site of the parenchymal partition, just like the expression of pSTAT3 (Figure 9j). This strong expression of Reg3β near the parenchymal partition was realized only in the liver lobe where the portal venous flow was preserved (RML in the ALPPS group). However, in the lobe with portal vein ligation like LLL in the PiLL group, inductions of Reg3α and Reg3β proteins were not observed at all, even near the partition site (Figure 9k,l).

## 3. Discussion

The basic research related to ALPPS was first published by Schlegel et al. [10]. They and others using animal models have stated that rapid liver hypertrophy in ALPPS is attributable to the increase of inflammatory cytokines—such as IL-6 and TNF-α—resulting from liver partition [10,11,12,13]. In line with those reports, production of IL-6 and TNF-α in the ALPPS group was significantly higher than in the PVL group. However, one noticeable result was that despite the RML/BW ratio being significantly less increased in the PiLL group than in the ALPPS group, IL-6 and TNF-α increased to as much as that in the ALPPS group. If rapid liver hypertrophy in ALPPS were simply attributable to the increase of inflammatory cytokines, the RML/BW ratio in the PiLL group should be comparable to that in the ALPPS group. Of course, we agree with the importance of paying notice to the tremendous increase of IL-6 in the ALPPS and PiLL groups, but a 30-fold increase in IL-6 over steady state might not necessarily be more important in initiating or accelerating liver regeneration than an 8-fold increase. There is even a possibility that the effect of an increase in IL-6 on liver regenration is already saturated with an 8-fold increase produced by PVL alone. We also confirmed the expression of *HGF* on day 1 as a key molecule in the lower stream of the IL-6 cascade, and its expression pattern was completely similar to that of IL-6. This fact again suggests that the excessive increase of *HGF* on day 1 in response to a massive IL-6 storm generated in the PiLL group does not take effect on additionally facilitating liver regeneration over the PVL group. The discrepancy between non-acceleration of liver hypertrophy in the PiLL group over the PVL group and the increased inflammatory cytokines as much as in the ALPPS group—including resulting HGF induction—strongly implies that there must be something else intervening in the secret mechanism underlying ALPPS.

Activation of the JAK/STAT3 pathway by IL-6 is known to accelerate cell proliferation. STAT3 plays an important role in liver regeneration after partial hepatectomy [18], and IL-6 deficient mice do not realize normal regeneration [19,20]. JAKs are activated through autophosphorylation, and in turn, they phosphorylate and activate STATs, which translocate into the nucleus to regulate gene transcription. Suppression of the JAK2/STAT3 pathway by JAK2 inhibitor resulted in the abolishment of the additional portion of liver hypertrophy in the ALPPS group, and the magnitude of volume increase reduced to basal hypertrophy, as obtained by the PVL or PiLL group. This result definitely indicates that the additional portion of liver hypertrophy in the ALPPS group is JAK2/STAT3-dependent. What is strange here is that the basal hypertrophy in the PVL or PiLL group was not affected by the JAK2/STAT3 pathway. Regarding this question, multiple studies with solid as well as blood tumors indicated that STAT3 and STAT5 were constitutively activated by tyrosine kinases other than JAK [21,22], and the JAK family protein consisted of four proteins, JAK1, JAK2, JAK3 and tyrosine kinase 2. Therefore, basal hypertrophy seen in the PVL or PiLL group might be dependent on STAT activation through a non-JAK pathway, or at least other than the JAK2/STAT3 pathway. In other words, our study happened to exhume that the molecular mechanism underlying liver regeneration by PVL also has not been fully elucidated. Further elucidation of a mechanistic pathway around IL-6/STAT in relation to PVL is intriguing.

Reg family proteins were discovered from the pancreatic juice of rats with chemically induced experimental pancreatitis [23,24], and afterwards classified into four subtypes [25]. Growing evidence links Reg3α and Reg3β proteins to regeneration of exocrine and endocrine tissues [26]. The beneficial effect of Reg3α on acute liver failure was reported in mice [27]. Lieu et al. investigated the role of the human hepatocarcinoma-intestine-pancreas/pancreas-associated protein (HIP/PAP) (another name for Reg3β) in liver regeneration using *HIP/PAP*-transgenic mice [28]. They showed that Reg3β accelerated the accumulation/degradation of nuclear pSTAT3. Conformity of Reg3 expression and STAT3 activation in the liver tissue strongly suggests that expression of Reg3 family protein is attributable to the activation of the JAK2/STAT3 pathway in our ALPPS model as well.

Unlike the contradiction observed between IL-6 and the regeneration rate of the liver, expression profiles of *Reg3* genes were parallel to the regeneration rate and also to the PCNA labeling index. According to Loncle et al., Reg3β induced by IL-17 promoted cell growth through activation of the JAK2/STAT3-dependent pathway in pancreatitis [17]. Furthermore, they showed that adding recombinant Reg3β promoted cell proliferation in the pancreas cancer cell line. The mitosis-facilitating role of Reg3β was reported in other experiments as well. Simon et al. reported that Reg3β stimulated liver regeneration after partial hepatectomy through the protein kinase A signaling pathway [29]. These observations reinforce our hypothesis that Reg3β induced in the ALPPS group activated the JAK2/STAT3 pathway and resulted in rapid liver regeneration.

The limitation of the present study is that we could not fully elucidate the secret why the up-regulation of *Reg3* family genes was triggered in the ALPPS group only. However, we showed that the expression of Reg3β was limited in the liver tissue where the portal venous flow was maintained. As stated by Starzl, insulin is a principal, although not the only, portal hepatotrophic factor playing a pivotal role in liver regeneration [30]. Even under cytokine stimulation, in order to trigger the Reg3 up-regulation, adding some factor like insulin supply through portal venous flow might be required. In addition, deviations of the expression location closer to the partition site may suggest that the commitment of some autocrine or paracrine factors other than humoral factors might be involved. The reason for more intense expression of Reg3β in zone 3 remains elusive. Recently, however, there are reports showing the existence of central-vein-associated lineage-restricted progenitor cells responsible for liver regeneration [31,32]. The relationship with them is intriguing.

Finally, the anatomy—and probably some physiological and/or immunological aspects as well—of the liver are different between rats and humans. In addition, step 1 of the original ALPPS procedure includes complete devasculization—affecting not only the portal vein but also the hepatic artery—of segment IV. These differences may have additional effects on liver regeneration in a human setting. However, given that a clinically often-used modification of ALPPS in which the parenchymal partition is laid along the Cantlie line (our rat model is more compatible to this modified one than the original ALPPS) also induces faster liver regeneration than PVL, the mechanism of rapid liver regeneration newly elucidated by the present study will help further clarify the details of multifactorial mechanism underlying the ALPPS method.

## 4. Materials and Methods

### 4.1. Animals

Male Sprague-Dawley rats, 8 weeks old and weighing 250–300 g, were used (CLEA Japan, Tokyo, Japan). The animals were housed in chip-bedded cages in an air-conditioned room (24 ± 1 °C) with controlled 12-h light/dark cycles. They were allowed free access to water and standard rat chow, and received humane care under the protocol approved by Animal Research Committee of Akita University (No. a-1-3011).

### 4.2. Surgical Procedures and Study Design

Under anesthesia by intraperitoneal injection of medetomidine hydrochloride, midazolam and butorphanol tartrate, a midline laparotomy was laid. For PVL, portal vein branches to the caudate, left lateral, left median and right lobes were ligated with 7-0 nylon under an operating microscope (TOPCON, Tokyo, Japan). During this procedure, portal branches were carefully isolated and arterial branches were untouched so that the blood perfusion of the portal vein ligated lobes could be maintained. In the ALPPS group, a partition of the liver was placed from the liver surface up to just anterior to the inferior vena cava using a bipolar coagulation forceps. In the PiLL group, the liver partition was placed in the center of left lateral lobe. In the study by Schlegel et al., they created a remote injury model (kidney, spleen and lung injuries) to assess the effect of systemic inflammatory response syndrome-like conditions on ALPPS liver hypertrophy [10]. However, they did not measure the strength of induced inflammation by remote organ injury, including blood IL-6 or TNF-α level; and their model may fail to detect potentially responsible factors which might be produced by the partition of the liver itself. Therefore, we newly developed a PiLL model in which only the partition site was altered from ALPPS to avoid the flaw derived from injuring organs other than the liver. The abdomen was closed in layers. Rats were sacrificed to collect blood samples and liver tissues from RML on days 1, 3 and 7 (*n* = 6 for each group per each time point). Blood samples were collected from the heart and they were centrifuged at 4000 rpm for 5 min. Serum was stored at −80 °C until use. Aliquots of liver tissues of RML were frozen in liquid nitrogen and stored at −80 °C. The remaining liver tissues were fixed in 20% formaldehyde. The weight of the RML of the liver and BW were measured on days 1, 3 and 7. RML/BW ratio (%) was denoted as liver hypertrophy index. In addition to these groups, a sham operation group (*n* = 6) was created, in which only laparotomy was performed, and sacrificed the next day (day 1) to obtain the blood and liver tissue samples as a control. Another 6 rats were used to ascertain the RML/BW ratio without intervention.

### 4.3. Serum Liver Enzymes

Serum on days 1, 3 and 7 was analyzed for AST, ALT and LDH levels using Transaminase CII Test Wako and Lactate dehydrogenase CII Test Wako (Wako Pure Chemical Industries, Osaka, Japan).

### 4.4. Enzyme-Linked Immunosorbent Assay (ELISA) for Serum Inflammatory Cytokines

IL-6 and TNF-α were measured in the serum on day 1 with ELISA kits (Rat IL-6; Immuno-Biological Laboratories, Gunma, Japan and Rat TNF-α; R&D Systems, Minneapolis, MN, USA).

### 4.5. Immunohistochemistry

Liver tissues fixed in formaldehyde were embedded in paraffin and sectioned into 3μm thickness. After deparaffinization, antigen retrieval and the removal of endogenous peroxidase, the sections were incubated overnight at 4 °C with a mouse monoclonal anti-PCNA antibody (1:200 dilution; Dako, Glostrup, Denmark), anti-phospho histone H3 antibody (1:500 dilution; Proteintech Group, Chicago, IL, USA), anti-Reg3α antibody (1:50 dilution; R&D Systems), anti-Reg3β antibody (1:50 dilution; R&D Systems) or a rabbit monoclonal anti-phosphorylated STAT3 antibody (1:200 dilution; Cell Signaling Technology, Danvers, MA, USA), followed by an incubation for 30 min at room temperature with TaKaRa POD Conjugate For Tissue (Takara Bio, Shiga, Japan) for visualization. The sections were immersed in diaminobenzidine solution and counterstained with hematoxylin.

To evaluate hepatocyte proliferation on days 1 and 3, the average percentage of PCNA-positive cells to total hepatocytes in 10 random high-power fields was established as the PCNA labeling index.

### 4.6. JAK2 Inhibitor in ALPPS Model

To explore the role of the JAK2/STAT3 pathway, selective JAK2 inhibitor (AG490, Chemscene, Monmouth Junction, NJ, USA) was used. Another set of animals was used for JAK2 inhibition experiments. We prepared again three groups of PVL, ALPPS and PiLL for AG490 administration as well as their corresponding control groups (*n* = 6 for each group). AG490 (10 mg/kg) was administered intraperitoneally 2 h before the operation with 0.5 mL/kg of dimethylsulfoxide (DMSO) as a vehicle. In the control groups, only DMSO was administered. All rats of this experiment were sacrificed on day 1 to obtain liver samples.

### 4.7. Quantitative Real-Time Reverse Transcription Polymerase Chain Reaction (RT-PCR)

Total ribonucleic acid (RNA) was extracted from the liver tissues on day 1 using RNeasy Mini Kit (QIAGEN, Venlo, the Netherlands). Total RNA of 1 μg was reverse-transcribed into complementary deoxyribonucleic acid (cDNA) using PrimeScript 1st strand cDNA Synthesis Kit (Takara Bio). To quantify mRNA expressions of *IL-6*, *TNF-α*, *HGF*, *Reg3α*, *Reg3β* and *PCNA*, quantitative real-time PCR (RT-PCR) was performed with LightCycler 480 SYBR Green I Master (Roche, Basel, Switzerland). Results of RT-PCR were expressed as the relative expression to *GAPDH*. The primer sequences for real-time RT-PCR are shown in Table 2.

### 4.8. cRNA Microarray for Gene Expression Profiling

In the PVL and ALPPS groups, gene expression was profiled by a cRNA microarray system using liver tissues of RML (a hypothetical part for FLR) obtained from three rats on day 1. Total RNA was extracted with TRIzol Reagent (Invitrogen, Carlsbad, CA, USA) and purified by SV Total RNA Isolation System (Promega, Madison, WI, USA). RNA samples were quantified by an ND-1000 spectrophotometer (NanoDrop Technologies, Wilmington, DE, USA) and the quality was confirmed with an Experion System (Bio-Rad, Hercules, CA, USA). The cRNA was generated from total RNA of 100 ng and labeled, using GeneChip™ WT PLUS Kit and hybridized to GeneChip™ Rat Gene 2.0 ST Array (Applied Biosystems, Waltham, MA, USA). All hybridized microarrays were scanned by an Affymetrix scanner. Relative hybridization intensities and background hybridization values were calculated using Affymetrix Expression Console™ (Affymetrix, Santa Clara, CA, USA).

Raw signal intensities of all samples were normalized by quantile algorithm with Affymetrix Power Tool version 1.15.0 software (Affymetrix), then were applied to Linear Models for Microarray Analysis (limma) package of Bioconductor software [33,34]. We established the criteria for regulated genes: (up-regulated genes) limma *p*-value < 0.05 and ratio ≥ 2.0-fold, (down-regulated genes) limma *p*-value < 0.05 and ratio ≤ 0.5-fold.

### 4.9. Statistical Analysis

Data were expressed as mean ± standard deviation. Statistical analysis was performed by unpaired Student’s *t*-test. Statistical significance was defined as *p* < 0.05.

## 5. Conclusions

An increase in inflammatory cytokines like IL-6 due to liver parenchymal partition alone was not enough to produce the ALPPS effect. The JAK2/STAT3 pathway seems to play a crucial role in the additional increase of the liver volume in ALPPS over PVL, but not in the basal hypertrophy produced by PVL alone. Activation of the JAK2/STAT3 pathway in ALPPS was independent of inflammatory cytokines. Expression of Reg3α and Reg3β in the liver tissue remnant was specific to ALPPS and could play a significant role in rapid liver hypertrophy in ALPPS in conjunction with an activation of the JAK2/STAT3 pathway.

## Figures and Tables

**Figure 1 ijms-21-04077-f001:**
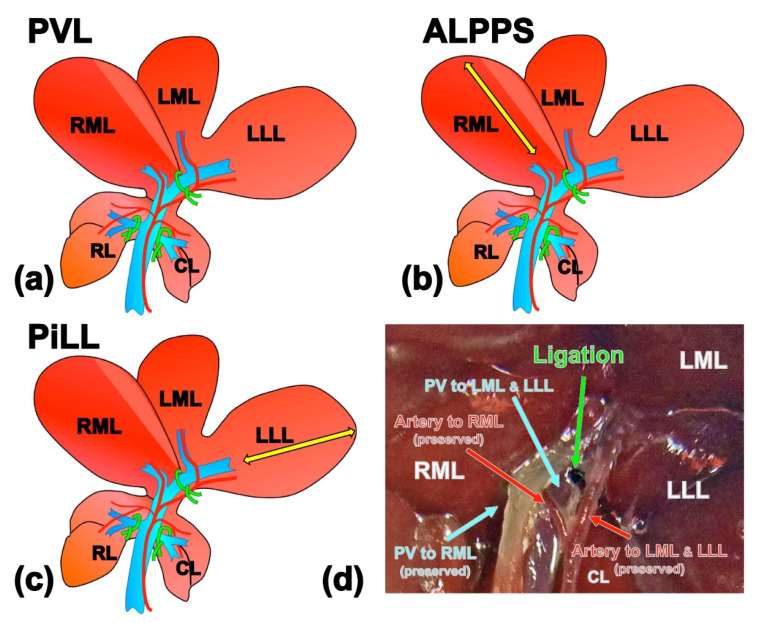
Illustrations of experimental models. (**a**) Portal vein ligation (PVL) group, portal branches to the caudate lobe, left lateral and left median lobes and right lobe were ligated, which accounted for 80% PVL. (**b**) Associating liver partition and portal vein ligation for staged hepatectomy (ALPPS) group, the right median lobe was split along the right side of the demarcation line that emerged after ligating the left branch of the portal vein in addition to PVL. (**c**) Partition inside the portal vein Ligated Lobe (PiLL) group, liver partition was placed inside the left lateral lobe (portal vein ligated lobe) in addition to PVL. Arterial branches to all liver lobes were preserved. (**d**) Demonstrative snapshot of PVL at the common branch to LML and LLL. Artery is conserved so that the perfusion is preserved even after PVL. Yellow arrows show the partitioning line. RML, right median lobe; LML, left median lobe; LLL, left lateral lobe; RL, right lobe; CL, caudate lobe; PV, portal vein.

**Figure 2 ijms-21-04077-f002:**
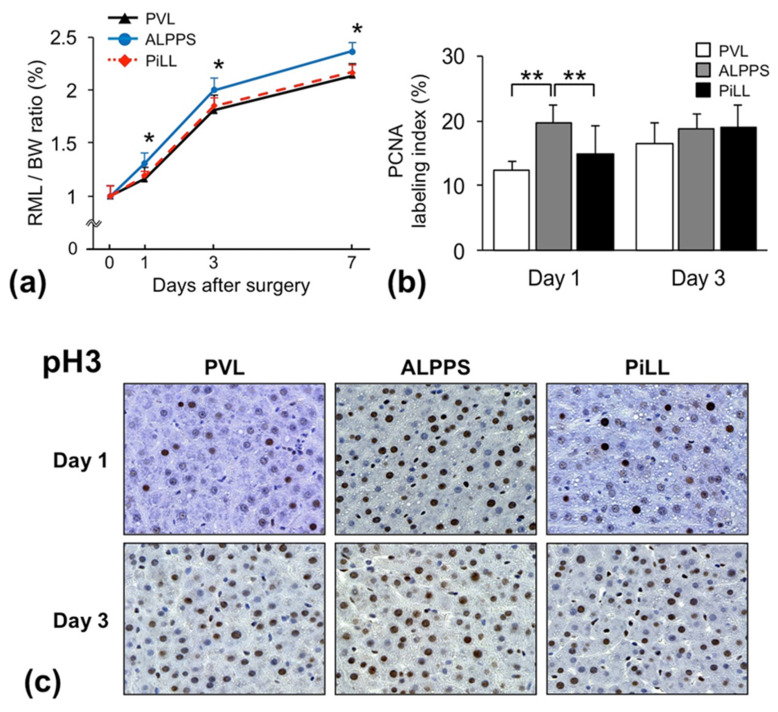
Chronological changes of liver hypertrophy, hepatocyte proliferation after surgery. (**a**) Weight of the right median lobe of the liver to body weight (RML/BW) ratio. (**b**) Proliferating cell nuclear antigen (PCNA) labeling index. (**c**) Immunohistochemistry for phosphorylated Histone H3 (pH3) on days 1 and 3. Values are expressed as mean ± standard deviation (SD); *n* = 6 for each group at each sacrifice time point; * *p* < 0.05 compared with the PVL and PiLL groups; ** *p* < 0.05.

**Figure 3 ijms-21-04077-f003:**
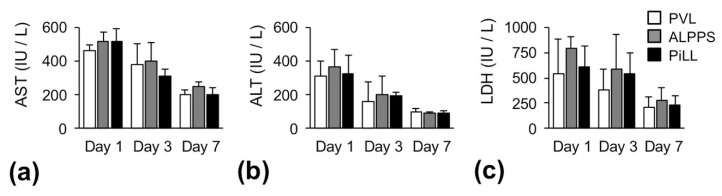
Chronological changes of serum liver enzymes after surgery. (**a**) Aspartate aminotransferase (AST). (**b**) Alanine aminotransferase (ALT). (**c**) Lactate dehydrogenase (LDH). Values are expressed as mean ± standard deviation (SD); *n* = 6 for each group at each sacrifice time point.

**Figure 4 ijms-21-04077-f004:**
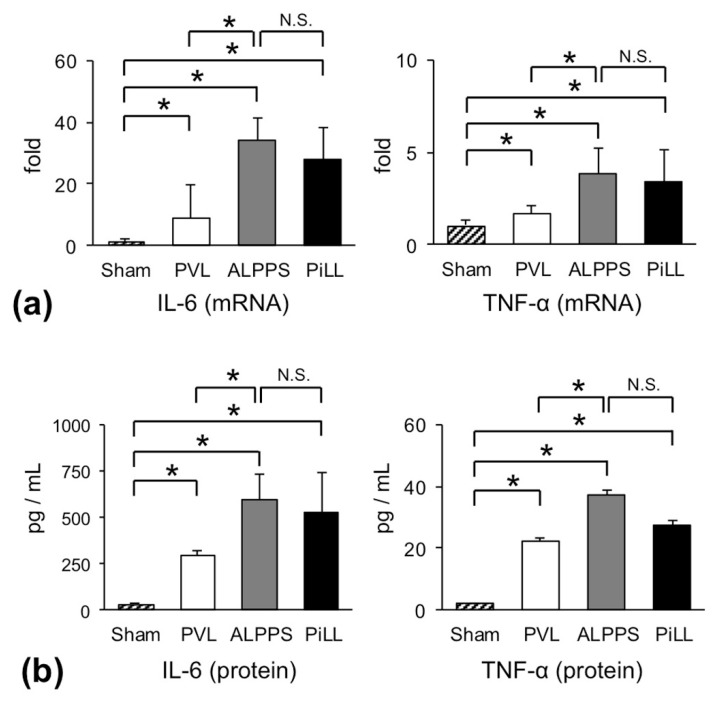
The levels of inflammatory cytokines on day 1. (**a**) Messenger RNA expressions of *interleukin* (*IL*)*-6* and *tumor necrosis factor* (*TNF*)*-α*. (**b**) Serum concentrations of IL-6 and TNF-α. Values are expressed as mean ± SD; *n* = 6 for each group; * *p* < 0.05; N.S., not significant.

**Figure 5 ijms-21-04077-f005:**
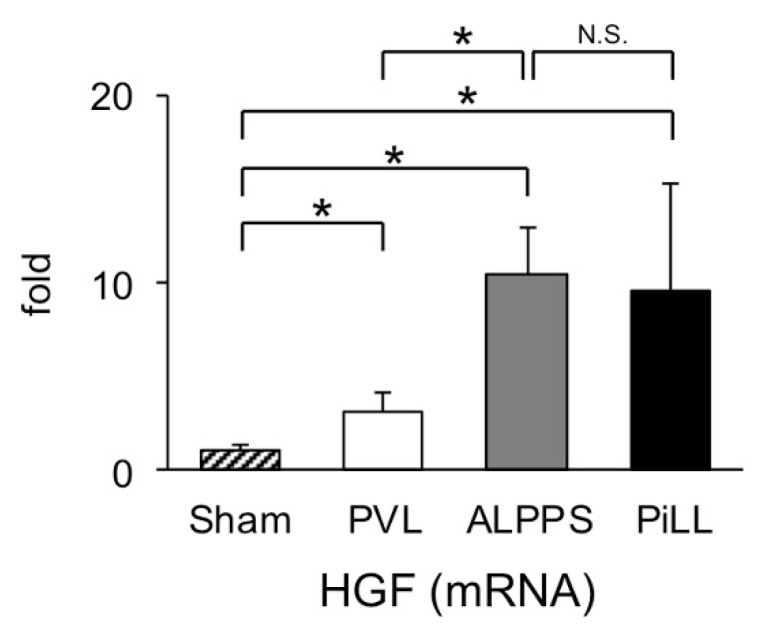
Expression of *hepatocyte growth factor* (*HGF*) in the future liver remnant (right median lobe) on day 1. Values are expressed as mean ± SD; *n* = 6 for each group; * *p* < 0.05; N.S., not significant.

**Figure 6 ijms-21-04077-f006:**
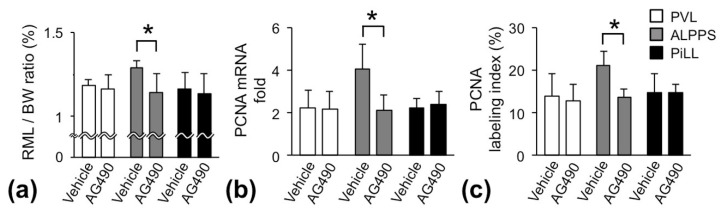
Blocking of the JAK2/STAT3 pathway by JAK2 inhibitor, AG490. (**a**) Weight of the right median lobe of the liver to body weight (RML/BW) ratio. (**b**) Messenger RNA expression of *proliferating cell nuclear antigen* (*PCNA*). (**c**) PCNA labeling index. Samples were taken on day 1 after administration of Janus kinase (JAK) 2 inhibitor AG490; values are expressed as mean ± SD; *n* = 6 for each group; * *p* < 0.05.

**Figure 7 ijms-21-04077-f007:**
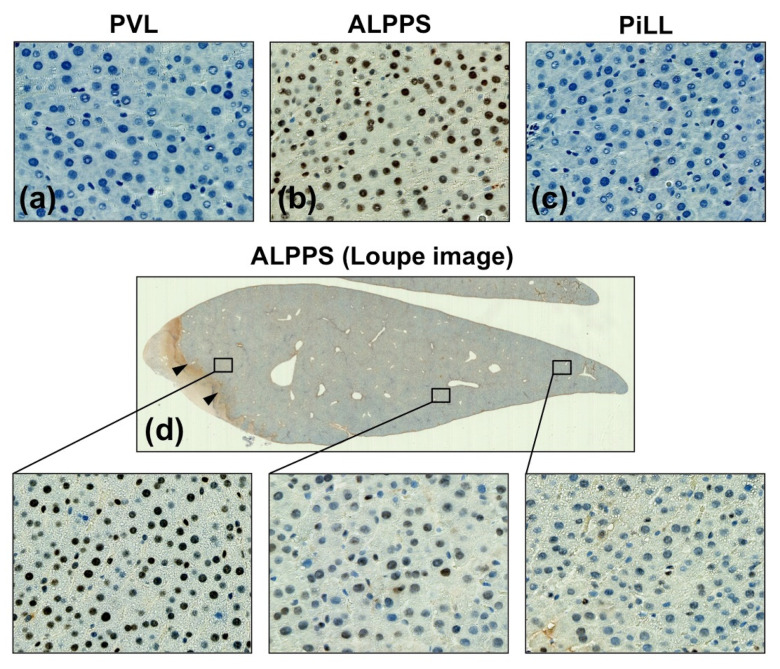
Immunohistochemistry for phosphorylated signal transducer and activator of transcription 3 (pSTAT3) on day 1. (**a**) PVL group, (**b**) ALPPS group, (**c**) PiLL group, (**d**) The expression was stronger near the partition site in the ALPPS group. Arrowheads show the partition site.

**Figure 8 ijms-21-04077-f008:**
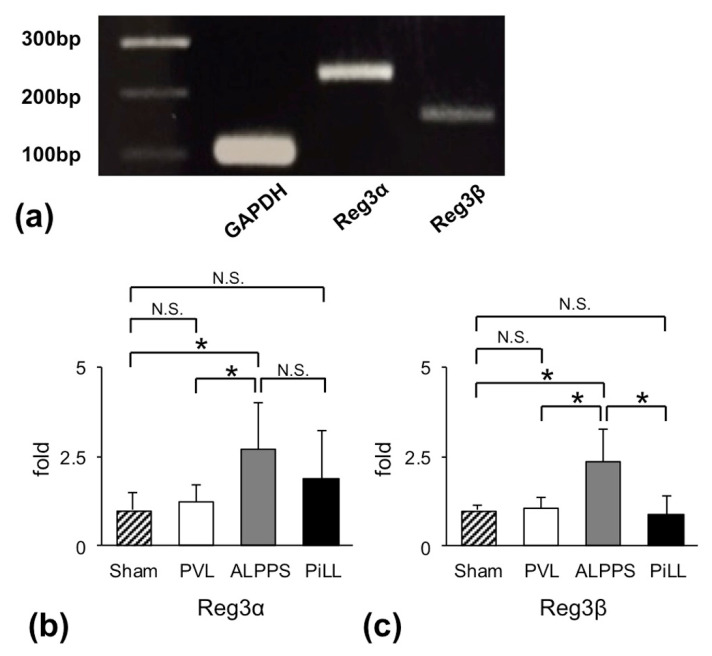
Messenger RNA expressions of *regenerating islet-derived* (*Reg*)*3α* and *Reg3β*. (**a**) Expressions in sham-operated liver. (**b**) Quantitative expressions of *Reg3α* messenger RNA on day 1. (**c**) Quantitative expressions of *Reg3β* messenger RNA on day 1. Values are expressed as mean ± SD; *n* = 6 for each group; * *p* < 0.05; N.S., not significant.

**Figure 9 ijms-21-04077-f009:**
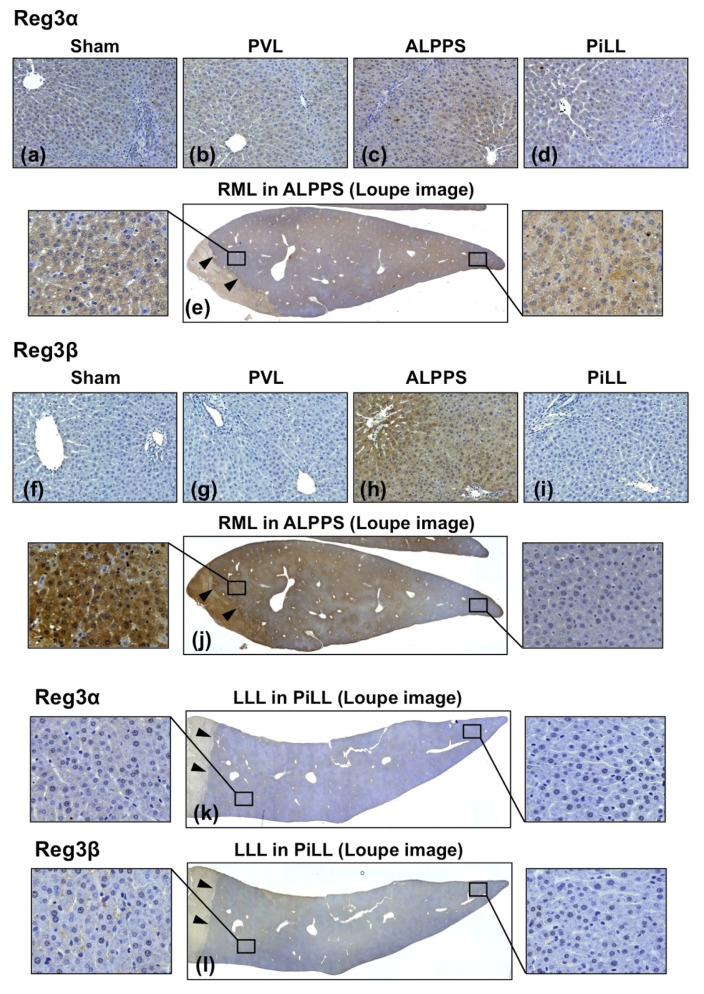
Immunohistochemistry for regenerating islet-derived (Reg)3α and Reg3β on day 1. (**a**–**d**) Reg3α in the right median lobe (RML)—future liver remnant—of each group. Although Reg3α staining was observed in every group, the staining intensity was enhanced only in the ALPPS group. (**e**) Loupe image of Reg3α staining in the ALPPS group. Reg3α was evenly expressed across the RML. (**f**–**i**) Reg3β in the RML of each group. Reg3β staining was negative in the sham, PVL and PiLL groups. In the ALPPS group, Reg3β was very strongly expressed especially around the central vein. (**j**) Loupe image of Reg3β staining in the ALPPS group. The staining was more intense in the tissue closer to the site of parenchymal partition. (**k**,**l**) Reg3α and Reg3β staining in the left lateral lobe, where the portal vein was ligated and the liver partition was placed, in the PiLL group. Neither Reg3α nor Reg3β was expressed. PVL, portal vein ligation group. ALPPS, associating liver partition and portal vein ligation for staged hepatectomy group. PiLL, partition inside the portal vein ligated lobe group. Loupe images are low magnification pictures of the liver tissue where the parenchymal partition was placed. Arrowheads show the partition site.

**Table 1 ijms-21-04077-t001:** Statistically up-regulated 104 genes in the ALPPS group compared to the PVL group.

Gene	Classification	Ratio	Gene	Classification	Ratio
*Akr1c2*	Alcohol metabolism	14.32	*Il7*	Immune response	2.044
*Adh4*	Alcohol metabolism	2.596	*Il1r1*	Immune response	2.025
*XAF1*	Apoptosis regulatory protein	2.259	*Dhrs7*	Metabolism	7.958
*Procr*	Blood coagulation	2.425	*St6galnac4*	Metabolism	2.505
*Pla2g7*	Blood coagulation	2.395	*Akr1c12*	Metabolism	2.338
*Itgb8*	Cell adhesion	4.272	*Ldhb*	Metabolism	2.057
*Siglec10*	Cell adhesion	2.108	*Dcps*	mRNA metabolism	2.187
*Ppdpf*	Cell growth/differentiation	2.021	*Ndnf*	Neuronal factor	6.066
*Reg3a*	Cell growth/differentiation	5.619	*Gfra1*	Neuronal factor	2.839
*Reg3b*	Cell growth/differentiation	3.748	*Slc1a2*	Neuronal factor	2.757
*Pla2g2a*	Cell growth/differentiation	3.803	*Ptprt*	Neuronal factor	2.313
*Ech1*	Energy metabolism	2.998	*Sarm1*	Neuronal factor	2.278
*Acsm5*	Energy metabolism	2.990	*Adora2b*	Neuronal factor	2.276
*Acsm2a*	Energy metabolism	2.847	*Kcnn2*	Neuronal factor	2.201
*Pltp*	Energy metabolism	2.688	*Asrgl1*	Neuronal factor	2.026
*Slc13a5*	Energy metabolism	2.649	*Asb15*	Protein metabolism	10.38
*Cxcl13*	Immune response	7.468	*Slc7a8*	Protein metabolism	2.374
*Il1b*	Immune response	6.861	*Glul*	Protein metabolism	2.059
*Siglec5*	Immune response	6.649	*Pcsk5*	Protein metabolism	2.037
*Clec4a2*	Immune response	5.488	*Pbsn*	Purin metabolism	2.098
*Il7r*	Immune response	3.849	*Pbsn*	Purin metabolism	2.015
*Cxcl1*	Immune response	3.438	*Rhbdf2*	Signal transduction	2.248
*Cish*	Immune response	3.361	*Sectm1b*	Signal transduction	2.185
*Ccl7*	Immune response	3.189	*Inhbe*	Signal transduction	2.093
*Il10*	Immune response	3.134	*Cyp2j4*	Stress response	2.553
*Ccl3*	Immune response	3.113	*Cyp3a9*	Stress response	2.521
*Adgre1*	Immune response	3.051	*Cybb*	Stress response	2.140
*Clec4a3*	Immune response	2.985	*Cyp4b1*	Stress response	5.069
*Slamf6*	Immune response	2.972	*Ephx2*	Stress response	3.787
*Il1a*	Immune response	2.917	*Serpina3m*	Stress response	3.509
*Csf2rb*	Immune response	2.714	*Ubd*	Stress response	3.229
*Timd4*	Immune response	2.630	*Hspb1*	Stress response	3.008
*Siglec8*	Immune response	2.618	*Cyp8b1*	Stress response	2.840
*Pilra*	Immune response	2.592	*Rac2*	Stress response	2.576
*Cd22*	Immune response	2.530	*Tbxas1*	Stress response	2.479
*Selp*	Immune response	2.462	*Steap4*	Stress response	2.474
*Cd7*	Immune response	2.396	*Nos1ap*	Stress response	2.437
*Cd14*	Immune response	2.372	*Reg3g*	Stress response	2.430
*Cxcr5*	Immune response	2.329	*Sult1c2*	Stress response	2.427
*Emr4*	Immune response	2.306	*Naip5*	Stress response	2.342
*Il2rg*	Immune response	2.283	*Cyp7b1*	Stress response	2.096
*Cd44*	Immune response	2.275	*Clec4a1*	Stress response	2.079
*Samsn1*	Immune response	2.250	*Rnd2*	Stress response	2.052
*Cd72*	Immune response	2.226	*Fxyd2*	Transport	2.290
*Igsf6*	Immune response	2.180	*Slco1a1*	Transport	2.139
*Cebpd*	Immune response	2.163	*Slc43a3*	Transport	2.137
*Csf1r*	Immune response	2.160	*Slc13a3*	Transport	2.110
*Fcgr2a*	Immune response	2.118	*Snx10*	Transport	2.110
*Slamf7*	Immune response	2.093	*Slc22a8*	Transport	2.009
*Cd37*	Immune response	2.090	*Fam169b*	Unknown	3.655
*Cd84*	Immune response	2.083	*RT1-N2*	Unknown	2.759
*Tcp11l2*	Immune response	2.080	*Klra7*	Unknown	2.749

**Table 2 ijms-21-04077-t002:** Primer sequences for real-time RT-PCR.

Genes	Forward	Reverse	Size (bp)
*IL-6*	TGCTCTGGTCTTCTGGAGTTC	TGTTGCTCAGACTCTCCCTTC	249
*TNF-α*	ATGTACCTGGGAGGAGTCTTC	AGAGTAATGGGGGTCAGAGTC	188
*HGF*	AAGAGTGGCATCAAGTGCCAG	CTGGATTGCTTGTGAAACACC	145
*Reg3α*	TGACTGAAGCTGAATGAAAGG	CAAGCAAGTACAGCCTTGTCATG	236
*Reg3β*	CTCCCTCACAGTTAAGATGTTGC	CCTAACTGCCACATGAGACTTTC	165
*PCNA*	TATTTGGCTCCCAAGATCGAAG	TTGGTGACAGAAAAGACCTCAG	121
*GAPDH*	ACATCAAGAAGGTGGTGAAGC	ATGGGAGTTGCTGTTGAAGTC	104

IL, interleukin; TNF, tumor necrosis factor; HGF, hepatocyte growth factor; Reg, regenerating islet-derived; PCNA, proliferating cell nuclear antigen; GAPDH, glyceraldehyde-3-phosphate dehydrogenase.

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
