# Peer review of "Reg3α and Reg3β Expressions Followed by JAK2/STAT3 Activation Play a Pivotal Role in the Acceleration of Liver Hypertrophy in a Rat ALPPS Model"

_ijms, 2020, doi:10.3390/ijms21114077_

Round 1
Reviewer 1 Report
In this report, the authors created a new model of 80% of ALPPS based on the split of the lateral left lobe (which is also a deportalized lobe), and studied the molecular episodes which may drive the first events of liver regeneration independent of the context of pro-inflammatory stimulus. They observed that, although there is a significant increase of IL-6 and TNF-a in ALPPS in comparison with PVL group, with a significant increase of hepatocyte proliferation on day one post-surgery. They also observed that blocking the JAK/STAT pathway, the ALPPS proliferation rate diminished to levels similar to PVL-PiLL groups, while in these two groups proliferation rate did not change. Additionally, and on basis on a comprehensive cRNA microarray system, the observed that two proteins Reg3a and Reg3b was constitutively expressed in ALPPS group, but not in the rest of groups. IHC studies on liver sections confirmed this issue. Authors conclude that there is a regenerative stimulus not driven by the pro-inflammatory IL-6/JAK/STAT pathway, and in which Reg3a and Reg3b may develop a role.
Under the context of liver regenerative procedures for resection of bilobar metastatic disease, ALPPS is considered as the most extreme procedure to induce a regenerative stimulus on the FLR (about 7-10 days). This extreme regenerative stimulus is not driven by a one single event, but there are multiple mechanisms which act simultaneously to induce such stimulus. There are two possibly theories which may explain it: the hemodynamic (ALPPS avoids collateral irrigation to the deportalized lobe) and the humoral (the damage induced by the procedure increase the humoral response on the FLR). Many mechanisms have been proposed, but no one has be demonstrated as the “gold standard” for the regenerative procedure under ALPPS context. Of course, stimulus induced by inflammatory cytokines are important, but I agree with authors that there is not enough to induce such dramatic increase of the FLR volume in such amount of time. In this sense, this paper propose an alternative pathway based on Reg proteins expression, which may act as an IL-6 independent activators of the JAK/STAT pathway. This contribution is interesting because enlighten alternative pathways by which ALPPS-driven liver regeneration takes place.
But, although the topic is interesting, there are major issues that need to be arranged prior to consider this report for publication.
Major points:
- The study is focused in the first events of the stage 1 (PVL and split of the FLR) of the ALPPS. Therefore, formerly is not a complete “ALPPS” procedure because there is a lack of the stage 2 (hepatectomy). Although the molecular events that drives regeneration in both stages seem to be different (in stage 2 the regenerative stimulus is lower than stage 1), is more accurate to consider this as first stage of ALPPS.
- To establish the proliferative index, authors used PCNA and p-histone H3 immunolabelling. Why they did not use Ki67 protein?. This protein cover all stages of cell proliferation.
- It is well-known that there is a close relationship of IL-6 and HGF expression during the first stages of liver regeneration, not only in experimental models but in human reports too. Did the authors established the levels of expression of HGF in the FLR of the different groups?. It a good indicator of the IL-6 activity in hepatocytes.
- The authors created a new model by splitting one of the deportalized lobes (named as PiLL). Although it is not the real situation (in human the split is always performed on the FLR), the results of the model (in terms of increase of the FLR) are the same as a PVL, although IL6 or TNF levels are increase (but not similar or higher than ALPPS). This result suggest that the effect of the split is more local than systemic. Based on the immunohistochemical expression of STAT-3 (Fig. 6) and Reg3b, reinforce the hypothesis, and this response and depend of the activation of the resident Kupffer cells or HSC near the split zone. Regarding the PiLL model, did the authors studied the volume of this lobe during the procedures? Was any alteration of PCNA index although the deprivation of portal flow? What were the IL-6 and TNF-a levels on this lobe?.
Reviewer 2 Report
This study gives new insight into mechanisms of liver regeneration triggered by ALPPS technique in a rat model. The study is well performed and the results are conclusive and well discussed.
Comment:
Since the anatomy of the liver - and probably also some physiological and immunological aspects - differ between rat and humans, it remains not completely clear that the same factors acutally underly liver regeneration after ALPPS in the human system. In fact, one of the major reasons to perform parenchmal dissection during the first stage of ALPPS is to (completely: artery and portal vein) devascularize segment IV of the liver. This may introduce some extent of necrosis of segment IV which may have additional effects on liver regeneration. This difference should be mentioned in the discussion.
Author Response
Thank you very much for precious suggestion.
Here is our reply to the comments by the reviewer.
- Since the anatomy of the liver - and probably also some physiological and immunological aspects - differ between rat and humans, it remains not completely clear that the same factors acutally underly liver regeneration after ALPPS in the human system. In humans, complete devasculization of segment IV is performed during the step 1 which may introduce some extent of necrosis. It may have additional effects on liver regeneration. This difference should be mentioned in the discussion.
Replay:
According to your suggestion, we added the following paragraph at the end of discussion.
LINE 316: Finally, the anatomy—probably some physiological and/or immunological aspects as well—of the liver is different between rats and humans. In addition, step 1 procedure of the original ALPPS includes complete devasculization—not only the portal vein but also the hepatic artery—of segment IV. These differences may have additional effects on liver regeneration in a human setting. However, given that clinically often-used modification of ALPPS, in which parenchymal partition is laid along the Cantlie line—our rat model is more compatible to this modified one than the original ALPPS—, also induces faster liver regeneration than PVL, the mechanism of rapid liver regeneration newly elucidated by the present study will help further clarify the details of multifactorial mechanism underlying the ALPPS method.
Round 2
Reviewer 1 Report
All my questions have been properly answered. Thank you. Thus, I recommend this manuscript to be published in its present form.